The circadian calling activity of a lebinthine cricket with high-frequency calls is unaffected by cicada choruses in the day

http://orcid.org/0000-0002-4324-6305 Tan Ming Kai 1 orthoptera.mingkai@gmail.com
http://orcid.org/0000-0002-2177-9549 Robillard Tony 1
http://orcid.org/0000-0002-7870-760X ter Hofstede Hannah 2 3
1 Institut de Systématique, Evolution, Biodiversité (ISYEB), Muséum National d’Histoire Naturelle , CNRS, SU, EPHE, UA, Paris , France
2 Dartmouth College, Department of Biological Sciences , New Hampshire , United States of America
3 Graduate Program in Ecology, Evolution, Environment and Society, Dartmouth College , New Hampshire , United States of America
González-Morales Nicanor
Electronic publication date: 2023 Jan 12
Publication date: 2023
Volume: 11
Electronic Location ID: e14641
Received 2022 Sep 1; Accepted 2022 Dec 5
Copyright: © 2023 Tan et al.
Copyright year: 2023
Copyright holder: Tan et al.
License: This is an open access article distributed under the terms of the Creative Commons Attribution License, which permits unrestricted use, distribution, reproduction and adaptation in any medium and for any purpose provided that it is properly attributed. For attribution, the original author(s), title, publication source (PeerJ) and either DOI or URL of the article must be cited.
License URL: https://creativecommons.org/licenses/by/4.0/

Keywords: Acoustic niche partitioning, Coexistence, Diurnal, Lebinthini, Southeast Asia

Funding: Fyssen Foundation Postdoctoral Fellowship Wildlife Reserves Singapore Conservation Fund Ming Kai Tan’s work was supported by the Fyssen Foundation Postdoctoral Fellowship and the Wildlife Reserves Singapore Conservation Fund. The funders had no role in study design, data collection and analysis, decision to publish, or preparation of the manuscript.

==============================
Background

Many factors can influence circadian rhythms in animals. For acoustically communicating species, both abiotic cues (such as light and temperature) and biotic cues (such as the activity of other animals), can influence the timing of signalling activity. Here we compare the 24-h singing activity of the cricket Lebinthus luae in the laboratory and field to assess whether the presence of other singing insects influences circadian rhythm.

Methods

Acoustic monitors were placed in four localities in Singapore and the number of L. luae calls were counted for 10 min of each hour. Individuals from the same localities were captured and recorded in the laboratory in silence but with similar abiotic conditions (temperature and light cycle) as they experience in the field, and the number of calls over 24 h was quantified.

Results

The 24-h pattern of L. luae singing was not significantly different between laboratory and field recordings. Singing activity peaked in the morning, with a secondary peak in the afternoon and a smaller peak at night. In the field, L. luae sang in the same locations and at the same time as diurnally singing cicadas, suggesting that the sympatric cicada chorus did not affect the circadian rhythm of communication in this species. Acoustic niche partitioning could potentially explain the ability of this cricket to call alongside cicadas: L. luae sings at higher frequencies than sympatric cicadas, unlike nocturnally singing cricket species that overlap with cicadas in frequency.

Introduction

Acoustic signalling for mate attraction and territory defence is common across animals, and signals often follow predictable circadian rhythms that reflect the ecology of each species (Schmidt & Balakrishnan, 2015; Rivas, da Bauzer & Meireles-Filho, 2016). Many factors can influence the circadian rhythm of acoustic signalling, such as changes in light and temperature (Clink et al., 2021; Feng & Bass, 2016), endogenous time-keeping mechanisms (Fergus & Shaw, 2013), social cues from conspecifics (Favreau et al., 2009; Fuchikawa et al., 2016), the presence of other acoustically signalling species (Hart et al., 2021), or a combination of these factors. When co-occurring species have similar circadian rhythms, the Acoustic Niche Hypothesis (ANH) predicts that selection will favour differences in signalling behaviour to help reduce errors in recognition and avoid acoustic interference (Krause, 1993; Farina, 2014). These differences may include producing sounds at different frequencies (Krishnan, 2019; Allen-Ankins & Schwarzkopf, 2022) or avoiding temporal overlap on short time scales (Hart et al., 2021). Community-wide partitioning of acoustic space in time and frequency suggests long-term adaptation to local acoustic environment (Farina, 2014). Little is known, however, about whether similar acoustic signalling circadian rhythms between species are due to a common response to abiotic factors, such as light and temperature, or could be synchronised by direct acoustic interactions between co-occurring species.

Tropical forests, due to their high biodiversity, tend to have more severe competition for acoustic niche-space than temperate habitats (Hart et al., 2021). In many Southeast Asian forests, the choruses of crickets and cicadas are among the most ubiquitous and conspicuous sounds. Ground crickets and cicadas in these forests tend to call at similar low frequencies (peak frequency <10 kHz) (Riede, 1996, 1997). These choruses are usually separated in time, with crickets being mainly nocturnal and cicadas mainly diurnal. There are, however, exceptions to these patterns in song carrier frequency and circadian rhythm. Cricket species in the tribe Lebinthini (Gryllidae: Eneopterinae) differ from other crickets by producing high-frequency (some even ultrasonic) calls (e.g., Robillard & Desutter-Grandcolas, 2004; Tan et al., 2021). Based on 24-h recordings in the laboratory, many of these lebinthine crickets appear to be diurnal or have complex acoustic circadian rhythms that peak during the day and night (Tan & Robillard, 2021).

Unlike in their natural environment, crickets recorded in isolation in the laboratory are not affected by acoustic competitors (both conspecific and heterospecific) or eavesdropping predators. The patterns observed in artificial conditions are thus more likely to reflect the intrinsic circadian clock (Tan & Robillard, 2021) or abiotic factors. Therefore, by comparing patterns observed in the field and laboratory, we can screen for potential biotic factors, such as acoustic competitors, influencing the circadian rhythm in the natural environment. Although Riede (1996, 1997) investigated the acoustics interactions of crickets and cicadas in Southeast Asia, studies combining laboratory and field data have not been undertaken for eneopterines and other Southeast Asian crickets.

In this study, we focus on the lebinthine cricket species Lebinthus luae Robillard & Tan, 2013 from Singapore, which produces broad-spectrum calls with a peak frequency of 18 kHz (Robillard & Tan, 2013) and exhibits two diurnal and one nocturnal peaks in calling activity in the laboratory (Tan & Robillard, 2021). In forests where L. luae are found, other singing insects exist that could potentially interfere with crickets’ ability to communicate. At least two cicada species commence singing around sunrise, generating a chorus that coincides with the increase in calling activity of L. luae in the laboratory. There are also low-frequency calling cricket and katydid species that have been described as largely nocturnal. We had two goals for this study. First, we tested whether the circadian rhythm in L. luae calling activity differs significantly between data collected in the field and in the laboratory. Second, we investigated potential overlap in the timing and frequency of L. luae calls with cicada choruses (i.e., ensembles of cicadas calling at the same time) and calls of other species of crickets and katydids in the same localities, particularly focusing on the sunrise period.

Materials and Methods

Acoustic recordings in the field

Lebinthus luae occurs only in a few isolated patches of forests in Singapore, usually close to the coast (e.g., Tan, 2010, 2012, 2013; Tan, Ngiam & Ismail, 2012). To monitor the circadian rhythm of the calling activity of L. luae in Singapore, four isolated localities were studied: secondary hillside forest in Hindhede Nature Park (N1.3485, E103.7755), coastal forests in Labrador Nature Reserve (N1.2659, E103.8032) and Sentosa Island (i.e., Imbiah Nature Trail) (N1.2570, E103.8138) and secondary forest in the southern parts of Ubin Island (N1.4089, E103.9723) (Figs. 1A and 1B). Sampling of the four localities was limited to the inter-monsoon period between mid-September to early November 2021 to minimise the effect of varying rainfall and temperatures between the different recording periods and the effect of heavy rain during the monsoon period on the recordings. Permission for collecting material and setting up in-situ recorders was granted by the National Parks Board [NP/RP18-064-2a], Singapore.

Figure 1 Map of Singapore with the four localities (A), environment where L. luae are found (B), deployment of acoustic recorder (C), contrast in the circadian rhythms in the field and laboratory (D).

Map of Singapore showing the four localities in which the circadian rhythms of the calling activity of L. luae was studied (A). The environment in which L. luae are found (B). Deployment of a Wildlife Acoustics Song Meter (SM) Mini acoustic recorder in the field (C). Contrast in the circadian rhythms in the calling activity of L. luae in Singapore based on field data (green line) and data collected from isolated males in the laboratory (blue line) (D). The presence of negative values for the standardised number of echemes per hour is due to the standardisation so that the laboratory and field recordings were comparable. The green and blue polygons bordering the lines represent the 95% confidence intervals. Grey shade represents nighttime; white represents daytime; transitions show the average sunrise and sunset times over the study period. A GAMM with gaussian error structure was fitted, with R2 = 0.36. The P-value provided indicates that there were no significant differences in the circadian rhythms in the calling activity of L. luae based on field and laboratory data.

Wildlife Acoustics Song Meter (SM) Mini acoustic recorders were attached to tree trunks at knee height and programmed to record continuously for 48 h with a sampling frequency of 96 kilo-samples per second and channel gain of 18 dB (16 bit) (Fig. 1C). The microphone was placed at around 45° downward to better sample the forest floor where the crickets dwell. Two recorders were programmed to record two different parts of the site simultaneously at each time. The two recorders were placed at least 20 m apart to ensure that they were not recording the same individual cricket(s) at the same time.

After 48 h (hence two consecutive circadian cycles), sound data were retrieved from both recorders and recordings were repeated in two other parts of the locality for the subsequent 48 h. Recorders were never placed in the same position within each locality. In total, six repeats of 48-h recordings were done for each recorder. Hence, a total of 24 circadian cycles of sound data were obtained (six repeats × two circadian cycles per repeat × two recorders) for each locality. After accounting for failed recordings (e.g., incorrect settings, problems with SD cards), 88 circadian cycles were analysed and used to feed the models. Sunrise and sunset times for each day were obtained from Meteorological Service Singapore (2022). The ambient temperature was recorded using a HOBO 8K Pendant® Temperature logger (model: UA-001-08; Onset, Bourne, MA, USA): 28.9 ± 1.6 °C.

Acoustic recordings in the laboratory

Wild individuals were collected from the same four localities for recording in isolation under standardised laboratory conditions. Recordings of individuals from Hindhede Nature Park, Labrador Nature Reserve and Pulau Ubin were already recorded and published in Tan & Robillard (2021). Individuals were collected from Sentosa and were recorded to supplement the laboratory data using the same methodology described in Tan & Robillard (2021). Briefly, individuals were recorded for 24 h in a sound attenuating box with a 11:13 h light/dark cycle (similar to the photoperiod in the field). The ambient temperature was recorded using a HOBO 8K Pendant® Temperature logger (model: UA-001-08; Onset, Bourne, MA, USA): 30.3 ± 2.5 °C. In total, 37 male individuals from the four localities were used for modelling (Hindhede n = 8; Labrador n = 11; Ubin n = 9; Sentosa n = 9).

Acoustic analyses of recordings

The calling song of L. luae consists of two parts: the first part corresponds to a number of well-spaced and short syllables denoted as “clicks”; and the second part is a trill of ca. 1.0 s in duration and made up of a number of syllables set closer together (Robillard & Tan, 2013). The trill was more consistent and unmistakably more noticeable on the spectrogram than the series of clicks (which can be confused with noise). To determine the level of calling activity in each hour of the circadian cycle, we counted the number of trills of L. luae (i.e., number of complete echemes (a first-order assemblage of syllables; Baker & Chesmore, 2020) in a standardised 10-min sampling window between 48 to 58 min of each hour in the circadian cycle. We manually looked through the spectrograms and identified trills of L. luae using Raven Lite 2.0.0. Because very faint trills may be confused with non-cricket sounds, and because they could potentially be calls of distant crickets detected by the other recorder, we only counted the number of trills with a threshold of average Power Density >−65.0 dB FS between the band of 16 to 19 kHz. At this power density, it was possible to identify the trills of L. luae more accurately. Specifically, it allowed us to differentiate the numerous and distinct syllables within the trill part of L. luae calls from the broad-band spectrum calls with similar frequency and duration ranges observed in some sympatric katydid species (e.g., Phyllomimus inversus and Mecopoda sp.). Additionally, to determine the level of calling activity during sunrise, we also counted the number of trills of L. luae at 5-min intervals between 06:00 and 07:30 of each circadian cycle.

To assess whether singing activity of cricket and cicada species overlap in time, we extracted the start times of cicada choruses using Raven Lite 2.0.0. We also extracted the calling activity of other low-frequency calling crickets. This was done only when crickets were near enough to the recorder to produce identifiable signals (with a threshold of average Power Density >−65.0 dB FS). When many crickets were singing at the same time, it was difficult to identify individual signals because their songs have a similar dominant frequency. Therefore, information about the calling activity of these crickets were supplemented with personal field observations. We measured the frequency spectra of the cicada in field recordings from the power spectrum in Raven Lite 2.0.0.

Statistical analyses

All statistical analyses were conducted with R version 4.1.3 (R Development Core Team, 2022). To avoid biases caused by raining, we excluded sampling windows in which raining occurred when modelling the circadian rhythm of the crickets’ call activity. A generalised additive mixed effects model (GAMM) with gaussian error structure for the standardised number of echemes per hour (see below) was fitted using the function ‘gam’ from the R package ‘mgcv’ (Wood & Wood, 2015). As the raw values obtained in the field and laboratory are not comparable (e.g., multiple individuals in the field vs. an isolated individual in the laboratory), a standardised number of echemes per hour was used instead. To make the data more comparable, we only used the field data from the first day of each deployment to make the number of circadian rhythms more comparable (i.e., 44 circadian cycles in the field vs. 37 circadian cycles (i.e., individuals) recorded in the laboratory). Furthermore, standardised values of echemes per hour for each dataset (i.e., field vs. laboratory) were obtained by subtracting the mean from each value and dividing by the standard deviation. The two datasets were combined and a similar standardisation was performed for the combined dataset.

To test whether the circadian rhythm of L. luae differed significantly between the field and laboratory recordings, we obtained smooth lines to visualise the circadian rhythm of the call activity, in that the time of day in intervals of 1 h (HR) was fitted as a smoothed fixed effect using cyclic cubic smoothing splines used as the smoothing method. The cyclic cubic smoothing splines were used to ensure that there was no discontinuity between 23:59 and 00:00 of the following day. A significant effect for this smoothed fixed effect indicates that there is a distinct circadian rhythm for either or both data source(s). To compare the circadian rhythms between data collected in the field and isolated individuals recorded in the laboratory, an interaction of the time of day in intervals of 1 h and dataset (laboratory or field) was fitted as a smoothed random effect. A significant result for this smoothed random effect indicates that the circadian rhythm is significantly different between the two data sources. To account for autocorrelations, a number unique for each circadian cycle and locality was used as a random intercept. Temperature, after scaling about its means (TEMP), was fitted as a non-smoothed fixed effect.

We also obtained smooth lines to visualise the L. luae calling activity around the sunrise period, a GAMM with gaussian error structure for the standardised number of echemes at 5-min intervals was fitted. This time interval was fitted as a smoothed fixed effects with thin plate regression spline. To compare the calling activity around sunrise period between data collected in the field and isolated individuals recorded in the laboratory, an interaction of the time of day in intervals of 5 min and dataset (laboratory or field) was fitted as a smoothed random effect. To account for autocorrelations, a number unique for each circadian cycle and locality was used as a random intercept. Temperature (TEMP) was also fitted as a non-smoothed fixed effect. For this model, we only used the field recordings and excluded recordings in which no trill over the 60-min sampling window.

To investigate potential overlap temporal overlap between L. luae calls with cicada choruses around sunrise, we identified the start and end times of the cicada choruses on smooth line showing L. luae calling activity. To investigate potential overlap in the dominant frequencies of L. luae calls with cicada choruses, we used a Kruskal-Wallis rank sum test, followed by a posthoc pairwise Mann–Whitney U-tests with Bonferroni correction to compare the dominant frequencies of the cicada choruses and L. luae.

Results

Over the circadian cycle, the model (adjusted R2 = 0.36, n = 81 circadian cycles, Table 1) did not have a significant interaction between time of day (HR) and data source (F-value = 0.0, P-value = 0.640), indicating that calling activity circadian rhythm does not differ significantly between data collected in the field and in the laboratory. In both datasets, three peaks were observed (Fig. 1D). The highest peak was in the morning (around 08:00), a smaller peak occurred in the afternoon (around 15:00), and another small peak was seen at night. For the field data (n = 44 circadian cycles), an average of 20 ± 1 echemes were detected in the morning peak (08:00), 7 ± 0 echemes in the afternoon peak (15:00) and 1 ± 0 echeme in the night peak (23:00 and 0:00). For laboratory data (n = 37 circadian cycles), an average of 7 ± 0 echemes per individual were detected in the morning peak (07:00), 5 ± 0 echemes per individual in the afternoon peak (14:00) and 5 ± 0 echemes per individual in the night peak.

Table 1 Summary of GAMMs with gaussian error structure.

The standardised number of echemes per hour was fitted for Model 1 on the circadian cycle and the standardised number of echemes at 5-min intervals was fitted for Model 2 on the sunrise period. *P < 0.05; **P < 0.01; ***P < 0.001.

	Estimate ± SE	t-value	P-value	
MODEL 1: Circadian cycle				
Intercept	−0.02 ± 0.06	−0.26	0.793	
Data source	0.02 ± 0.09	0.27	0.785	
TEMP	0.03 ± 0.05	0.70	0.483	
		F-value	P-value	
Smoothed (HR): field data		35.4	<0.001***	
Smoothed (HR): lab data		7.10	0.047*	
Smoothed (data source, HR)		0.00	0.640	
Smoothed (random)		4.52	<0.001***	
MODEL 2: Sunrise period				
Intercept	0.22 ± 0.11	1.94	0.053	
Data source	−0.34 ± 0.16	−1.21	0.034*	
TEMP	−0.09 ± 0.06	−1.50	0.135	
		F-value	P-value	
Smoothed time: field data		47.9	<0.001***	
Smoothed time: lab data		9.09	<0.001***	
Smoothed (data source, time)		0.05	0.016*	
Smoothed (random)		8.54	<0.001***	

Around the sunrise period, the model (adjusted R2 = 0.53, n = 73 circadian cycles, Table 1) indicated a moderate but significant interaction between time of day and data source (F-value = 0.054, P-value = 0.016), indicating that calling activity around sunrise slightly differs between data collected in the field and in the laboratory (Fig. 2E). This can be attributed to the earlier morning peak observed in laboratory than in field data, as indicated by the line in Fig. 1D.

Figure 2 Calls, calling activity patterns and the frequency partitioning of crickets and cicadas in Singapore.

Calls and calling activity patterns of crickets and cicadas in Singapore. One hour spectrogram showing the transition from nocturnally singing crickets (band at ~6 kHz) to the “cicada ^” dawn chorus and the diurnally singing Purana sp. cicada and Lebinthus luae cricket (A). Insets (B–E) show short timescale spectrograms of the calls of the most common singing insects in the recording. The wav.file for this 1 h spectrogram is available as Data S1. Calling activity of L. luae around sunrise based on field data (green line) and data collected from isolated males in the laboratory (blue line) (F). The presence of negative values for the standardised number of echemes per hour is due to the standardisation so that the laboratory and field recordings were comparable. The polygons bordering the line represent the 95% confidence interval. Grey shade represents night-time; white shade represents daytime. A GAMM with gaussian error structure was fitted, with R2 = 0.53. Boxplots comparing the frequencies of the fourth harmonics of the cicada choruses and the dominant frequency of L. luae calls (G). Thick horizontal bar shows the median; lower and upper margin of the box indicate the inter-quartile range and whiskers refer to minimum and maximum data points.

Our field recordings contained cicada choruses starting in the morning (Fig. 2A) with calls of Purana sp. (Fig. 2D) commencing on average at 06:52 (SD = 13 min, earliest = 06:34, latest = 07:39) and calling throughout the day. Additionally an unidentified species that we call “cicada ∧” sang for an average of 12 ± 5 min at dawn, on average between 06:45 (SD = 6 min, earliest = 06:37, latest = 07:03) and 07:01 (SD = 7 min, earliest = 06:49, latest = 07.19) (Fig. 2C). The cicada choruses produced almost continuous sound due to their long calls and multiple individuals singing at the same time (Fig. 2A). Therefore, L. luae calls were almost always overlapping with cicada calls even on short time scales (Fig. 2E). Unlike L. luae, other sympatric cricket species (e.g., Gymnogryllus spp., Velarifictorus aspersus, and Ornebius spp.) call at lower frequencies (3–8 kHz) and overlap with the first and second harmonics of the cicada choruses (Fig. 2B). These crickets generally stop calling around the same time as the cicada choruses start in the morning (on average 07:08, earliest = 05:21, latest = 08:27) (Table S1). In contrast, the calling activity of L. luae increased during this period (adjusted R2 = 0.87, n = 72 circadian cycles, Fig. 2B, Table 1).

The dominant frequencies of the cicada species did not overlap with the dominant frequency of L. luae. For “cicada ^” and Purana sp., the fourth harmonics (15.1 ± 0.3 kHz and 12.0 ± 0.2 kHz, respectively) have the highest energy that are also closest to the spectrum of L. luae calls. The first to third harmonics of these cicadas are below 10 kHz. Following a Kruskal-Wallis rank sum test (χ2 value = 85.2, d.f. = 2, P-value < 0.001, n = 29 “cicada ∧”, 33 Purana sp., 35 L. luae), pairwise Mann–Whitney U-tests showed that (all pairwise P-values < 0.001 after Bonferroni correction) the dominant frequency of L. luae does not overlap significantly with those of the fourth harmonics of the two cicada choruses (Fig. 2G).

Discussion

We found that the circadian rhythm of calling activity does not generally differ significantly between crickets recorded in the field or in the lab. This result validates the idea that recording eneopterine crickets in isolation under artificial laboratory conditions can still offer meaningful information about their natural circadian rhythms. This also suggests that the observed pattern is predominantly influenced by the individual’s intrinsic biological clock (Fergus & Shaw, 2013) and abiotic environmental cues such as light (Clink et al., 2021; Feng & Bass, 2016), rather than social cues from conspecifics (Favreau et al., 2009; Fuchikawa et al., 2016) or the presence of other acoustically signalling species (Hart et al., 2021).

The nocturnal peak, however, appears to be marginally greater in the laboratory than in the field data. We postulate two plausible explanations for this difference. First, in the wild, L. luae have to contend with predation risks, such as nocturnal palm civets (Fung, Tan & Sivasothi, 2018) and reptiles (M. K. Tan, 2021, personal observations), and they might reduce calling activity in response to predator cues (e.g., ter Hofstede & Fullard, 2008). Second, couples might form during daytime calling with mating occurring more at night (M. K. Tan, 2021, personal observations), but under isolated laboratory conditions in the absence of females, males continue to sing. These hypotheses could be tested using playbacks of potential predator sounds in the presence or absence of females.

The second main finding of this study is that L. luae calls alongside the daytime cicada chorus, unlike low-frequency calling crickets that partition their calling activity with cicadas by calling predominately after dusk when the cicada choruses have stopped (Riede, 1996, 1997). By comparing circadian rhythms of calling in the laboratory and field, our results suggest that daytime calling activity by L. luae is not drastically affected by cicada choruses. Frequency partitioning, as predicted by ANH, may have allowed L. luae, with high-frequency calls (peak frequency = 18 kHz), to call alongside cicada choruses of slightly lower frequencies (maximum frequency of cicada choruses are 16 kHz) (Fig. 2). By avoiding frequency interference with cicadas, L. luae may still communicate acoustically with conspecifics during the day without having to segregate their calling time.

The divergence and diversification of the crickets from the tribe Lebinthini have been attributed to the clade’s novel communication system, including shifting to calls with high or ultrasonic frequencies (ter Hofstede et al., 2015; Benavides-Lopez, Ter Hofstede & Robillard, 2020; Tan et al., 2021). This divergence also coincided with the origin of echolocation in bat ancestors, implying that the novel communication system in the lebinthines can be a form of adaptation to avoid detection and predation by bats at night (Hill et al., 2021; Tan et al., 2021). Based on our study, we may offer another plausible hypothesis, that is, lebinthines can avoid frequency jamming with cicada choruses in the day owing to the shift to high-frequency calls in accordance with ANH.

Conclusions

Here, we show that the circadian rhythm of L. luae calling activity observed in the laboratory is representative of that in the natural environment. This suggests that the individual’s intrinsic biological clock and environmental cues play an important role in the circadian patterns, more so than influences from other acoustically signalling animals. This can be further supported by our observations that the calling activity of L. luae does not appear to be affected by the commencement of cicada choruses after sunrise. These crickets and cicadas can call at the same time because the dominant frequency of the L. luae call does not overlap with the maximum frequency of the cicada calls, and hence the crickets can still communicate acoustically without being masked by the cicada choruses. Further investigations into the circadian rhythm of lebinthine crickets and cicadas, including playback experiments and comparative studies that account for phylogeny, can offer more direct evidence for ANH in this system and further insights into the interactions.

Supplemental Information

Supplemental Information 1 R Codes.

Click here for additional data file.

Supplemental Information 2 Description of the calling activity and peak frequency of some sympatric low-frequency calling crickets and katydids.

+1 refers to the next following day.

Click here for additional data file.

Supplemental Information 3 Sound file obtained from Sentosa Island on 16 September 2021 around sunrise period containing the “cicada ^” dawn chorus.

Click here for additional data file.

Supplemental Information 4 Sound file obtained from Sentosa Island on 16 September 2021 around sunrise period containing the calls of nocturnal crickets.

Click here for additional data file.

Supplemental Information 5 Sound file obtained from Sentosa Island on 16 September 2021 around sunrise period containing the calls of L. luae and Purana sp. cicada.

Click here for additional data file.

Additional Information and Declarations

Competing Interests

Author Contributions

Field Study Permissions

Data Availability

Tony Robillard is an Academic Editor for PeerJ.

Ming Kai Tan conceived and designed the experiments, performed the experiments, analyzed the data, prepared figures and/or tables, authored or reviewed drafts of the article, and approved the final draft.

Tony Robillard conceived and designed the experiments, analyzed the data, authored or reviewed drafts of the article, and approved the final draft.

Hannah ter Hofstede analyzed the data, prepared figures and/or tables, authored or reviewed drafts of the article, and approved the final draft.

The following information was supplied relating to field study approvals (i.e., approving body and any reference numbers):

Permission for collecting material and setting up in-situ recorders were granted by the National Parks Board [NP/RP18-064-2a], Singapore.

The following information was supplied regarding data availability:

The code is available in the Supplemental File.

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
