# Peer review of "The circadian calling activity of a lebinthine cricket with high-frequency calls is unaffected by cicada choruses in the day"

_PeerJ, doi:10.7717/peerj.14641_

## Round 0.1 · original submission · Minor Revisions

· Academic Editor

Minor Revisions

I am sorry for the long time it took to provide a response. It was hard to obtain reviewers, I understand it is a common current problem.

I enjoyed reading your work. The reviewers highlight a number of minor concerns, please address them. Reviewer 3 mentions a problem with self-citation, I don't see a problem with self-citations. However, add other people's work to balance the references. Reviewers 1 and 2 ask for more experimental details and clarifications on the text. Those should be included as well.

Reviewer 1 ·

Basic reporting

No comment

Experimental design

The second question of this work is whether morning peaks of L. luae are affected by the commencement of cicada chorus after sunrise. The term "affected" is not clear, and leaves the question open to speculation. Does it mean that the cricket peaks can occur at different times in the field because of the presence of the cicadas, or that they have a different pattern? By saying only, that the general calling pattern in the 24 hrs is not significantly different between lab and field, does not automatically imply that the peaks in the morning are not affected. In fact, small differences of 10 or 15 minutes would not be found significant given the adopted model that tend to consider a 24 hr pattern and do not explain differences at a small temporal scale. On the contrary, a more specifically dedicated statistical analysis is required to respond to the second aim of the article, which conversely, in the present state, is not sufficiently addressed. Or maybe, this is not what the authors meant. At any rate, I invite them to reformulate the second aim with a clearer statement and, in case, to make a statistical analysis specifically addressing the morning time.
In the GAMM analysis, the authors omitted the “locality” effect. I think that it would be interesting to know if there were or were not significant differences between the 4 sites. In fact, differences due to local (abiotic and biotic) effects could be found between localities. By using a nested approach, it would be possible to solve this question.

Validity of the findings

An important issue is the lack of an analysis of the temporal overlap on short time scales between cicadas and Lebinthus luae. The last conclusion of the article is: "These crickets and cicadas can call at the same time because the dominant frequency of the L. luae call does not overlap with the maximum frequency of the cicada calls, and hence the crickets can still communicate acoustically without being masked by the cicada choruses". This means that the frequency plays a key role. However, to assess this statement it would be important to analyze whether cicadas and crickets overlap their calls. Otherwise, the final statement remains a mere hypothesis. On the contrary, a deeper analysis of the cicadas and crickets’ calls can confirm (or reject) this hypothesis and validate the statement. If this analysis in not possible then the final statement cannot be accepted because we do not know if the two species can actually overlap on short time scales and thus if the different dominant frequency play a role.
Similarly, in the results it is written that cicadas sing all day long and that night crickets stop calling as soon as cicadas start. It is not clear if the authors actually measured these data or they got them from the available literature. In the first case, they should report, at least in a table, some statistics (even descriptive) about the observed activity of these other species. In the second case, they should report the reference and move the statement in the discussion.

Additional comments

Was there a special reason to choose the inter-monsoon period to make the registrations? Is it the only period where the two species co-occur and sing?
Is 11:13 hour light/dark used in the lab approx. the photoperiod in the studied area?
Values of lab conditions (T and HR) are not indicated.
The value standardization makes the graph (Fig. 1) reporting negative values in the Y-axis. This is not actually possible. This creates the paradox that night peaks are lower than 0. I invite the authors to make a graph with not standardize values, otherwise the readers cannot see anywhere in the text, the actual values (in terms of mean and SD) of the observed data.

Reviewer 2 ·

Basic reporting

Dear all,
This is a well conducted and clear work within the field of bioacoustics. Even if the manuscript does not present major flaws, I would suggest publishing after minor revisions and as short communication.
The protocol is clear and well-designed, but the result section seems rather meager, as only a limited amount of data is presented. The latter is sufficient to support the hypothesis of an “internal clock” in the studied species, whilst other assertations resemble more speculations, especially since values regarding the number of cicada and cricket choruses (i.e., percentages, means, occurrence...) are not present within the manuscript or as Suppl. Material. Lastly, the number of replicates (in the laboratory) are rather low, decreasing the power of the statistical analyses and, as a consequence, of the results.
At any rate, I truly appreciated the methods, especially the protocol for analyzing and choosing the signals to be used in the analyses (i.e., from line 137 to 144). This could help other researchers dealing with long and complicated sound files recorded from field sites.

Experimental design

- Please, provide more details regarding the analyzed signals: percentages of cicada and cricket choruses per recording or day, occurrence, duration, etc. Considering that you are stating that cicada choruses do not affect the cricket calling activity, it would be useful for the reader to better understand what you define as “cicada chorus” (i.e., is a cicada sufficient? They always sing in group?) and how many choruses occurred simultaneously with the cricket songs within the analyzed time windows. The figures, in my opinion, are not sufficient to support your conclusions.
- Provide a clearer picture of the ecology and ethology of the studied species within the introduction (i.e., a few lines regarding the structure and timing of the cicada and cricket songs and for how long they sympatrically occur (a few months/weeks?).
- Provide the coordinates of the localities in the materials and methods. I suggest adding a map (either within the manuscript or as supplementary material).

Validity of the findings

- Within the conclusion, you state “These crickets and cicadas can call at the same time because the dominant frequency of the L. luae call does not overlap with the maximum frequency of the cicada calls...”. However, you did not analyze the frequency of the two species in this work, making your assertion more a speculation. I suggest rephrasing as hypothesis or adding references to support your statement (i.e., line 216-221, a figure is not sufficient to assume that this is the frequency range of this cicada species).
- Overall, I feel that the data here presented are valuable but limited. Therefore, the authors should be more cautions when they support their hypotheses within the discussion and conclusion, especially when comparing the cricket and cicada signals and calling activities. I suggest highlighting that the work supports the authors’ hypotheses, but further research is needed to confirm them.

Additional comments

Figures.
- Could you add some stats about figure 1? For example, how many recording were used to create the lines? In y axis, we can see the n of echemes, but on what basis (n echemes/rec/hour...)? The caption indicates per hour, it is also valid for the y axis?
- Fig. 1. Could you add the p-value of the GAMM?
- Fig. 2. What do you mean with “composite” in the caption?
Clarity.
- Line 113. What do you mean with “failed”? Were they too noisy or the recorders did not work?
- Line 114-116. Here, you mention sunrise and sunset times taken from the station, but did you also download information regarding temperature and RH? Similar data are important, since these parameters are known to influence insects’ calling activity. Moreover, the comparison between laboratory and field recording should consider T and RH. From your model, it seems that you added T as factor (e.g., see line 173-174), but I could find neither how you measured that in the field nor the T in the lab. In other word, either you point out that T and RH were retrieved from field sites or you explain why you did not measure these parameters.
- Line 124. Could you explain why 24 hours instead of 48?
- Line 135. Please, define “echeme”.
The writing is clear and grammatically correct. Even so, I would suggest some corrections at lines:
- 26. Remove “in this species” (unnecessary).
- 31. Remove “for these individuals as well” (unnecessary).
- 35. Remove “intense” (data regarding cicada choruses are not present) and substitute “does” with “did” (consecutio temporis).
- 38. Remove “the” before “cicadas in frequency”.
- 43. Remove “these” (unnecessary).
- 70. Remove “both” (unnecessary).
- 103. Remove “tend to” (unnecessary).
- 95 and 108. I suggest using “thus” or “hence” instead of “=”.
- 109. Remove “continuous” (unnecessary).
- 111. Remove “of the two” (unnecessary).
- 157. Add comma after “i.e.”
- 159-160. The usage of “to scale” seems wrong. Please check whether “about” is used as you expected.
References
- 54-58. I suggest adding references in here as well.
- The reference in text says “Reide” L64, 216, whilst the reference list indicates “Riede”. Please, correct.
Supplementary material
- The excel sheet is not very helpful without an explanation or a legenda. Moreover, the data seem scarcely informative.
- I suggest adding a sound file as example for the reader (I was curious to hear cricket and cicada choruses of a tropical forest).
- I suggest adding more information regarding the analyzed songs within the recording (occurrence, means, medians...).
- A map of the sites is appreciated as well.

Annotated reviews are not available for download in order to protect the identity of reviewers who chose to remain anonymous.

Reviewer 3 ·

Basic reporting

The manuscript is written in competent and clear English. Nevertheless, please avoid using colloquial expressions, e.g. line 25: instead of „lab“, you should use a more professional term (i.e. „laboratory“) throughout the text.

Please avoid self-citation wherever possible. I am aware that Lebinthus luae has been studied by only a few scientists so far, but references to your own work should be used sparingly and only when directly relevant for the topic. For instance, in lines 90-91, you do not need six references for this simple statement, one or two are more than enough.

Lines 115-116: You should cite a webpage in accordance with the PeerJ instructions for authors (please see https://peerj.com/about/author-instructions/).

Lines 228-230: Please provide a reference for this statement.

Figure 1: In the figure caption you should clearly state which statistical metrics are represented by the line itself and which by the green/grey band surrounding the line (are those confidence intervals?). Also, please provide the basic metrics of the GAMM in the figure caption, e.g. p-value, so that the reader can immediately see if there are statistically significant differences in the calling activity shown in the figure. Please use another colour for the line representing field data, in order to make it more distinct from the line representing laboratory data, since dark green is very similar to black in the present figure. I also suggest representing daytime by plain white (instead of orange shade) in the figure, to avoid using too many different colours and thus improving the visual aspect and readability of the figure.

Experimental design

The first two paragraphs of the introduction are well-organized and provide a relevant overview of the subject, however they are followed by a very short paragraph insufficiently explaining the motivation behind the study. Please expand on this by providing more information on similar studies on this or other species, clearly emphasizing the knowledge gaps that are to be filled by your research. The only reference currently cited in this paragraph is a paper by the authors – I find it hard to believe there is no other relevant research (on other species or groups) that have looked into a similar matter and can thus provide motivation for your study.

In their present form, study aims are poorly defined, with no relevant information on how you would achieve them. For instance, in lines 82-85 you write that “By conducting field recordings, we aim to test whether… calling activity differs significantly between data collected in the field and in the laboratory”, but how can one test the difference between two “treatments” by performing one of the “treatments”? You should rather say that by comparing the data collected in the field and in the laboratory, you aim to achieve a certain aim that you need to define here. The same goes for the potential effect of cicada choruses – how do you plan to quantify and test this? The relevant information is again missing. Please rewrite the study aims in a more clear and concrete way.

Validity of the findings

Line 192: Please refer to the Figure 2 here. However, since the simple statement that the calling activity of L. luae increased in this period is not enough to prove that it is unaffected by cicada choruses, I suggest that you quantify this increase e.g. by testing the correlation between the increased calling activity of cicadas and L. luae using Spearman or Pearson correlation coefficient (whichever is more suitable for your data). In this way, you would be able to show that e.g. there is a positive correlation or at least an absence of a negative one between the calling activity of cicadas and L. luae, which means they are active in the same period and therefore not affected by signal masking. This would also improve your discussion on the relationships between crickets and cicadas (lines 214-223) since at present you can only draw conclusions indirectly, from the comparisons of L. luae recordings in the laboratory and in the field, with no direct correlations with cicada signalling.

237-239: You cannot claim there is a lack of a significant effect if no statistical tests have been performed. Please choose an adequate method to test this. I suggest calculating Spearman or Pearson correlation coefficient (please see above).

Lines 236-237: Please draw a concrete conclusion regarding the observed lack of differences between laboratory and field data. You only mention this result here, but then immediately move on to a next result (regarding cicada choruses), without drawing any conclusions. Thereby, please take care to avoid repeating the same statements in discussion and conclusions; in the former you should comment on specific findings, and then try to generalize (as far as your findings allow) in the latter.

Additional comments

Line 129: The information on dominant frequency is already given in the introduction (line 80). Please avoid repetitions.

Line 230: “also” is redundant in this sentence.

Line 232: “their” is redundant in this sentence.

---

## Round 0.2 · accepted · Accept

· Academic Editor

Accept

Hi Dr. Tan,

In my opinion, you have adequately addressed all the original reviewers' comments and concerns. One reviewer has some minor concerns and suggestions. I'd suggest that you address their two main concerns. Make sure that the number of replicates is clearly stated and that you include a brief explanation for why you consider that cicadas sing in choruses but not L.luae by adding a short reason. I'd also strongly suggest considering their suggested text changes for the final version of the manuscript.

Nicanor

Reviewer 1 ·

Basic reporting

I am satisfied with the answers and corrections provided by the Authors. In my opinion, the article is now ready to be published.

Experimental design

No comment

Validity of the findings

No comment

Additional comments

No comment

Reviewer 2 ·

Basic reporting

Dear all,
The revised manuscript has significantly improved, acquiring more relevance and statistical power. I thank the authors for following our suggestions, implementing the statistics, and adding the recordings within the SM. I truly appreciated the discussion as well. In the first round, I suggested publishing as short communication after revising it, whilst the current version deserves publication as original article. Even so, I suggest acceptance after minor revisions, which mainly concern the writing (there are redundant/unclear/complicate sentences, probably related to management of changes and revisions by different authors).
Dear authors, I have two main concerns:
1. I could not find the number of replicates used to perform the Kruskal-Wallis (cicadas vs crickets)
2. Why you consider that cicadas sing in choruses, whilst the same seems not occur in L. luae, even if in the field you were not quantifying the number of calling crickets? You repetitively stated that more individuals could call in the field. Then, why these are not considered choruses? Perhaps, a brief explanation for readers could help.

Below some sentences that need rewriting (please, consider that I wrote my version to help/as a suggestion, but you are free to accept or refuse, provided that you improve the readability of the highlighted sentences/lines):
- 36-37. I suggest emphasizing that this is only an hypothesis, since your data represent only the tip of the iceberg in supporting the ANP in this scenario. So, instead of “likely explains” I suggest “could potentially explain” or “could explain”. “likely” indicates probability, whilst here potentiality is more appropriate.
- 48. Delete “the” before “presence...”
- 112. Acronyms have meaning if they are used often throughout the manuscript. You may delete MS, considering that you use once within the work.
- 52. “These differences MAY...”
- 55. Delete “the” before “local...”
- 66. Delete “diurnal” before “cicada...”
- 71. Delete “the” before “day and night.”
- 75. Delete “the” before “patterns...”
- 86. Delete “the” before “forests...”
- 87. Delete “the” before “crickets’ ability...”
- 86-90. References of these two sentences are needed.
- 93. Add “i.e., within the brackets. Here, the authors introduced the definition of cicada choruses, which is also repeated afterward. Once is sufficient, please remove redundant clarifications (example, line 221). Moreover, the sentence in the introduction (L. 63) rose a question: the calling activity you recorded in the field could be defined as well a “chorus”? What is the difference with the cicada choruses you observed and recorded?
- 91. Simplify the sentence: “First, we tested ...”
- 120. Delete “the” before “sound data...”
- 121. Add a comma after “recording”
- 126. Unclear. I suggest writing “...cycles were analysed and USED TO FEED THE MODELS”
- 156. I suggest removing “also”
- 164. Unclear. I suggest simplifying by removing “presence” and (L. 165) “from the sound files”
- 166. Delete “the” before “crickets”
- 170. The subject is unclear, as well as the meaning of the sentence. “To assess whether these species overlap in frequency, which can interfere with intraspecific communication when calling at the same time, we measured the frequency spectra of the cicada in field recordings from the power spectrum in Raven Lite 2.0.0.”.
- The paragraph related to statistical analyses has many redundant or unclear sentences. I suggest rephrasing:
- 176-177. “...we excluded sampling windows in which raining occurred, when modelling the circadian rhythm of the crickets’ call activity.”
- 179. Delete the text within the brackets, replace with “see below” since you describe standardization after.
- 180 onward. I suggest simplifying as follows: “As the raw values obtained in the field and laboratory were not comparable (e.g., multiple individuals in the field vs. an isolated individual in the laboratory), we only used the field data from the first day of each deployment to make number of circadian rhythms more comparable (i.e., 44 circadian cycles in the field vs. 37 circadian cycles recorded in the laboratory). Furthermore, standardised values of echemes per hours for each dataset (i.e., field vs. laboratory) were obtained by subtracting the mean from each value and dividing by the standard deviation. The two datasets were combined and a similar standardisation was performed for the combined dataset.”
- 191. “differed”
- 193. “...activity, in that...”
- 194. Remove “used” and “the”
- 194. Add “the” before “cyclic cubic...”
- 195. Simplify, example: “were used to ensure that there was not discontinuity...”
- 206. Add “thus” after “sunrise period,”
- 208. Simplify after the period, example: “This time interval was fitted as a smoothed fixed effects with thin plate regression spline.”
- 214-216. Simplify, example: “For this model, we only used the field recordings and excluded recordings in which no trills were recorded within the 60-min time window.”
- 218. Perhaps incorrect. I suggest: “To investigate potential temporal overlap between the calls of L. luae and the cicada choruses...”. The next sentence is unclear to me: “...identify the start and end times of the cicada choruses on smooth line showing...”
- 228. I suggest rephrasing: “the model (...) did not showed a significant correlation between HR and the source of the recordings (i.e., field and laboratory) ...”
- 232-233. I suggest simplifying “For field data (n = 44 circadian cycles), ...”
- 235. Remove “the” before “laboratory data”
- 241. “...indicated a moderate but significant interaction between time of the day...”
- 242. “indicating that calling activity around sunrise slightly differs...”
- 244. Clarify: “... to the earlier morning peak observed in laboratory than in field data, as indicated by the line in Fig. 1D.”
Indeed, once the authors provide answer to the raised questions and improved the readability of the above-highlighted sentences, publication may proceed.
Kindest regards,
R2

Experimental design

no issues

Validity of the findings

No issues

Additional comments

See 1. Basic reporting